# Biomimetic Diatom Biosilica and Its Potential for Biomedical Applications and Prospects: A Review

**DOI:** 10.3390/ijms25042023

**Published:** 2024-02-07

**Authors:** Ki Ha Min, Dong Hyun Kim, Sol Youn, Seung Pil Pack

**Affiliations:** 1Institution of Industrial Technology, Korea University, Sejong 30019, Republic of Korea; alsrlgk@gmail.com; 2Department of Biotechnology and Bioinformatics, Korea University, Sejong 30019, Republic of Korea; jklehdgus@korea.ac.kr (D.H.K.); youseul0419@korea.ac.kr (S.Y.)

**Keywords:** diatom, diatomite, biosilica, bone regeneration, wound healing, drug delivery

## Abstract

Diatom biosilica is an important natural source of porous silica, with three-dimensional ordered and nanopatterned structures referred to as frustules. The unique features of diatom frustules, such as their high specific surface area, thermal stability, biocompatibility, and adaptable surface chemistry, render diatoms valuable materials for high value-added applications. These attributes make diatoms an exceptional cost-effective raw material for industrial use. The functionalization of diatom biosilica surface improves its biophysical properties and increases the potential applications. This review focuses on the potential uses of diatom biosilica including traditional approaches and recent progress in biomedical applications. Not only well-studied drug delivery systems but also promising uses on bone regeneration and wound healing are covered. Furthermore, considerable aspects and possible future directions for the use of diatom biosilica materials are proposed to develop biomedical applications and merit further exploration.

## 1. Introduction

Diatoms are unicellular algae ranging in size from 2 μm to 2 mm, and they are the representative phytoplankton commonly found on Earth [1]. Diatoms, which are unicellular and photosynthetic microalgae, are prevalent in freshwater, seawater, and wet soils [2]. They exhibit broad morphological diversity, with almost 110,000 species [3,4]. Since diatoms account for 20–25% of total primary biomass production on land and 40% of marine biomass production, they have played an important role in ecosystems as marine biomass producers for millions of years [5]. Unlike other marine microalgae, diatoms form amorphous hydrated silica frustules with an outer cell wall structure [6].

Hard mineralized shells persist even after cell death, and the accumulation of these shells in soil yields diatomaceous earth (DE) or diatomite [7]. DE is a naturally derived material formed from the remains of diatoms that grow and are deposited in oceans or lakes. DEs, also known as kieselguhr, are mineral deposits formed from diatomaceous algae [8,9]. Commercially exploited deposits are of a relatively recent age, beginning in the Miocene epoch. Older deposits have undergone tectonic processes that have altered the texture and crystalline phase of the mineral. Amorphous silica is a constituent of diatom frustules and forms the primary component of DE. Other materials, including metal oxides, clays, salts (primarily carbonates), and organic matter, may be present in variable quantities [8]. The impurity content of a deposit is determined by chemical precipitation and atmospheric contact, as well as the prevailing environmental conditions.

Since DE is not produced from a single species compared to diatom frustules, the morphology of the frustule including pore size and distribution, as well as thickness of the silica skeleton, varies [10]. DE applications are limited to uses in abrasives, moisture-absorbent, and eco-friendly filters, and there is no high value-added application [11]. Increasing research interest in the utilization of diatom biosilica in recent years stems from its desirable properties including chemical inertness, biocompatibility, high mechanical and thermal stability, low thermal conductivity, uniform porous structures a high specific surface area, adjustable pore volume, etc. [11]. Unlike the production of synthetic silica materials with a micro or nanoscale structure using an expensive process of conventional nanofabrication, diatom biosilica of high purity can be harvested in an enormous quantity without the need for substantial energy and material inputs. Therefore, biosilica represents an unlimited, readily available, natural, low-cost, and renewable material [12]. Moreover, biosilica production is immensely environmentally friendly as toxic waste is not generated and the consumption of energy is lower than that involved in the production of synthetic silica-based materials.

The aforementioned properties of diatom biosilica allow for applications in the biomedical field, including biopharmaceuticals, targeted drug delivery, catalysis, biophotonics, chemicals and biosensors, and cancer diagnosis and treatment [13]. Biosilica is also used in a wide range of applications in various research fields such as wastewater treatment, enzyme immobilization, bioremediation, micro and nanofabrication, filtration, microdevices, protein separation, cement additives, imaging, photoluminescence, and trace gas detection (Figure 1) [14]. The main strength for the use of diatom biosilica is the highly-porous morphology of the frustule structures, appropriate for new functional material designs.

Although diatom biosilica has received great attention, research and applications of biosilica need to be diversified. Diatom biosilica is highly versatile in shape and properties, allowing for various applications [11,12,13]. There have been several reviews of biomedical applications [13,15,16] and bone tissue engineering [17,18], with various studies using diatom biosilica morphology as templates and drug delivery systems.

This review also focuses on traditional approaches and advances in biomedical applications. In particular, the recent multidisciplinary applications of diatom biosilica in the fields of bone regeneration and wound healing applications are compiled thoroughly. It also covers considerable aspects and possible future directions that could be explored for the use of diatom biosilica materials in biomedical applications.

## 2. Silicon Uptake and Frustule Formation

Diatom cell walls, known as frustules, represent a remarkable example of three-dimensional (3D) structures found in nature [19]. Diatoms are classified in two groups based on frustule symmetry: centric diatoms that exhibit radial symmetry with respect to an axis that passes through the cell center, and pennate diatoms that present bipolar symmetry with a longitudinal axis that runs parallel to the plane of symmetry [20]. Although they differ in shape, diatom frustules are commonly bipartite structures formed by two overlapping valves referred to as thecae. The epitheca and hypotheca, which are connected by girdle bands or cingula, comprise the upper and lower parts, respectively. This structure is frequently compared to the form of a Petri dish or pill box [21]. Porous elements, such as the cribellum, cribrum, and foramen, exhibit a hierarchical organization, with diverse patterns and pore diameters ranging from nanometers to micrometers [22]. The biosilicification process mainly occurs in unicellular algae. They are the most important organisms that exhibit this process, both in terms of the number of silicified structures they can produce and the global production of biogenic silica [23].

Biosilicification occurs in specialized intracellular compartments referred to as silica deposition vesicles (SDVs). Silicon, which is obtained from the environment in a soluble form as orthosilicic acid (Si(OH)_4_), enters the cell cytoplasm via specific silicic acid transporter proteins. These proteins comprise 10 transmembrane domains and can directly bind Si(OH)_4_ to glutamine regions to facilitate its transportation into SDVs [24]. Inside the SDV reaction vessel, Si(OH)_4_ is transformed into a silica network (SiO_2_) by the combined function of long-chain polyamines, silaffin proteins, and polyanionic silacidin peptides [25]. Cultivating diatoms in large quantities is relatively simple; the cell culture process involves providing inorganic salts as nutrients and sunlight for growth [26]. Alternatively, a more affordable source of diatom silica is DE, which has formed over a million years through fossilization of deceased algae and is presently extracted. Additionally, several studies on the biomineralization process of silica nanofabrication by diatoms have prompted the development of synthetic routes to create unique silica-based materials with mild reaction conditions [27]. Proteins and other organic substances related to diatom biosilica have been thoroughly purified and analyzed. Specifically, silaffins and silacidins have been isolated from diatom cell walls and used to create silica nanostructures including spheres, rods, and platelets in artificial conditions [28].

## 3. Diatom Biosilica: Structure, Purification, and Surface Modification

Naturally occurring diatomite microparticles contain impurities such as clay, volcanic gas, terrigenous particles, and organic matter. The main sources of these impurities are the local environment and aging process. Therefore, the purification of diatoms is crucial to obtain pure silica microshells which can be utilized in various applications.

### 3.1. Unique 3D Structure of Diatoms

The unique features of diatoms include their attractive silica-based architecture, which can be genetically controlled [29]. Diatoms comprise a cell wall called the frustule. Each frustule or biogenic silica shell of different diatom species exhibits a distinct nanostructure [30] and is enhanced with features such as spines, ridges, channels, and pores [8] with enormous porosity [31]. The structure of diatoms varies with species and may include hexagonal, rod-shaped, and circular forms [10]. Additionally, each species of diatom has unique surface properties (Figure 2A). The two most important types of diatoms are centric diatoms, which exhibit radial symmetry, and pennate diatoms, which have an elongated structure. Elongated diatoms are highly sought after for their high porosity owing to specialized pore structures [32]. In diatoms, the upper segments are referred to as epitheca, while the lower segments are named hypotheca and shaped like a Petri dish (Figure 2B) [33].

### 3.2. Purification of Raw Diatom Biosilica

Aging conditions and the local environment represent sources of impurities present in the microparticles of naturally found diatoms, including organic substances, clay, terrigenous particles, and volcanic gas [34]. Figure 2A shows images of raw DE. To obtain diatom frustules in the purest form, which are referred to as “isolated silica micro-shell”, it is necessary to remove all of the present impurities. It was reported that pure diatom frustules can be obtained using a pulverizer for diatom crushing or a piranha solution to remove impurities [35]. These processes are used most widely amongst many methods to obtain pure diatom frustules. In industrial applications, maintaining a high SiO_2_ content and the structural integrity of diatom frustules represents the two most important requirements for the purification of biosilica structures [36]. Studies have reported various procedures for purifying frustules from diatoms, including acid cleaning and baking as primary techniques [37,38,39,40]. Nonetheless, few researchers have combined these methods for the extraction of diatom frustules. Owing to the higher amount of organic matter in live diatoms compared to that in DE, both contain various trace metal elements. Consequently, obtaining the desired SiO_2_ content in the final product via a single extraction process poses a significant challenge.

Of all of the discussed methods for separating and purifying diatom frustules, pulverization and piranha solution treatment are the most common methods (Table 1). The piranha solution acts as a potent oxidizing agent which effectively removes organic layers and exposes hydroxyl groups on the diatom surface [41]. This approach offers relatively greater advantages including cost-effectiveness, simple purification, rapid process, and an eco-friendly (no solvents used) nature [35]. This technique can also be used for the purification of water by extracting toxic metal ions. Figure 3B,C show the purified diatom frustules.

### 3.3. Surface Modification of Diatom Biosilica

Owing to the presence of SiO_2_ in DE that serves as building blocks, the surface properties of purified DE can be customized to create a novel category of bioengineered materials for use in biomedical applications [16]. Silica carries hydroxyl (OH) groups on its surface, and this property can be easily utilized for functionalizing the DE surface using established chemical modification techniques [48]. Consequently, the diatom surface can be suitably modified. Advances in structural development over the past decade have been made through various techniques primarily focusing on approaches created for artificial silica particles. These strategies involve the application of organic monolayers, polymers, proteins, and coating with metal and inorganic oxide layers [16]. The most common techniques involve the use of reactive silanol (SiOH) groups accessible on the diatom surface that can be readily functionalized with many reactive species, for example, -NH_2_, -COOH, -SH, and -CHO [48,49]. These provide reliable coupling points for the fixation of chemical or biological constituents such as enzymes, drugs, proteins, aptamers, antibodies, DNA, and sensing probes, among others [16].

Photoluminescent compounds or fluorescent diodes are commonly used as trackers, while active molecules including carbon monoxide-releasing molecules, nitric oxide-releasing molecules, and scavengers of reactive oxygen species (ROS) are used [14]. Two widely adopted approaches are used for immobilizing active biomolecules onto the surface of chemically modified DE biosilica: non-covalent interactions involving physical adsorption and other weak interactions along with covalent immobilization involving strong covalent binding [50]. The main issue with non-covalent binding, such as electrostatic interaction, is its dependence on solution conditions, such as the pH or changes in ionic strength, which results in decreased stability. Therefore, the covalent binding of biomolecules to the surface of diatoms is frequently used in real-life applications that require stability and reproducibility. Further, silver nanoparticles (AgNPs) have been synthesized on the surface of diatomite using green synthesis and a novel green synthetic agent, *Pinus koraiensis* pinecone extract. The resulting DE biosilica composite coated with AgNPs has been reported to be well dispersed. It is suggested that the DE biosilica functions as a dispersant, thereby mediating NP synthesis without aggregation. The use of DE biosilica provides advantages such as well-dispersed formation and enhanced properties of AgNPs in environmentally friendly NP production [48].

## 4. Biomedical Applications of Diatom Biosilica

The primary advantages of using purified diatom or diatomite biosilica for biomedical purposes are as follows: frustule modification for functionalization, biocompatibility, genetic transformation for protein immobilization, and high availability of silica-derived diatoms [13]. Synthetic porous silica (macro-, meso-, or microporous) has been extensively studied owing to its distinct physicochemical characteristics [51]. However, its applications are limited by expensive and time-consuming manufacturing processes that require the use of noxious solvents that may contaminate the end-products. Biosilica, which is naturally synthesized by unicellular diatoms, has the potential to serve as an advanced biomaterial for developing relatively safer and more effective therapeutics in the future. Diatom-derived biosilica requires low-cost synthesis processes [49], and it is characterized by chemical inertness, low or non-toxicity, thermal stability, and high availability [48,52]. Additionally, biosilica has been widely used in DDSs owing to its high durability and adaptability when compared to other substances. Frustules extracted from both live cultures and DE particles have been used as efficient drug carriers [53].

### 4.1. Bone Regeneration

Silicon plays a crucial role in the formation and maintenance of bones by enhancing the function of osteoblast cells and promoting mineralization. Bone deformation and long-bone abnormalities are often associated with silica/silicon deficiency [54]. DE, a natural deposit of diatom skeletons, is a cheap and abundant source of biogenic silica that can be used in regenerative medicine applications [55,56,57,58] (Table 2).

Purified *Thalassiosira weissflogii* diatom, which has been chemically modified, was previously conjugated with 2,2,6,6-tetramethylpiperidine-1-oxyl radical, a potent ROS. This composite was then used in the development of a DDS to support bone cell growth and tested with ciprofloxacin as the drug of choice. The experiment showed that the initial release of the drug was fast; however, the controlled release lasted for approximately seven days. This technique is advantageous in treating infections related to dental or orthopedic surgery, enhancing bone cell adhesion, promoting cell proliferation, and preventing inflammation [14,59]. Sodium alendronate (ALE) can be incorporated in vivo into the nanostructured biosilica shells of cultured *T. weissflogii*. Additionally, Na ALE-functionalized frustules can be obtained by using an acid oxidative process while still maintaining their mesoporosity. Furthermore, ALE-biosilica represents an efficient osteoconductive biomaterial substrate in vitro, and it stimulates tissue regeneration via osteoblast-like cells (SaOS-2) and bone marrow stem cells while suppressing proliferation of osteoclast-like cells (J774) [60]. The mentioned study examined the effects of two novel bioactive agents DE and polyhedral oligomeric silsesquioxanes on the characteristics of chitosan/Na-carboxymethylcellulose polymer blend scaffolds. The authors also compared the impact of silica reinforcements with that of silica-substituted nano-hydroxyapatite particles [61]. A novel multifunctional scaffold has been developed to facilitate the engineering of bone tissue via a co-electrospinning mechanism. The scaffold comprises fibers that are loaded with an antibiotic; they are created by combining poly(hydroxybutyrate-co-hydroxyvalerate)/poly(ε-caprolactone) (PHBV/PCL) and pullulan fibers containing DE, a pharmaceutical element. A previous study examined the release of cefuroxime axetil (CA) from DE and the scaffold; CA was loaded into DE, PHBV/PCL fibers, or both [62].

**Table 2 ijms-25-02023-t002:** Bone regeneration applications of diatom/diatomite biosilica.

Application	Type	Function	Type of Functionalization	Loading Material	Ref.
Composite	Diatom (*Thalassiosira weissflogii)*	Osteoactive material	Bisphosphonates	-	[60]
	Diatomite	Polyelectrolyte scaffold	Chitosan/Na-carboxymethylcellulose	-	[61]
	Diatomite	Chitosan membrane	-	-	[63]
	Diatomite	PHBV-PCL fibrous scaffold	-	Pullulan	[62]
	Diatomite	Chitosan composites	-	-	[64]
	Diatomite	Silk fibroin	-	-	[65]
	Diatomite	Collagen/chitosan/hydroxyapatite nanocomposite	-	-	[66]
Material loading	Diatomite	Chitosan composite	Polyethyleneimine	BMP-2	[67]
		PHBV-PCL fibrous scaffold	-	Melatonin	[68]
		Diatomite scaffold	-	Copper	[69]
Biocoating	Diatomite	Magnesium implants	-	ZrO_2_ particle	[70,71]
		Ceramic coating	-	-	[72]

PHBV-PCL, poly(3-hydroxybutyrate-co-3-hydroxyvalerate and poly(ε-caprolactone); BMP-2, bone morphogenetic protein 2.

Studies have reported the use of silica-based implants in bone regeneration. A new protective coating for orthopedic magnesium implants has been developed and its physicochemical properties were evaluated. The coating was synthesized using the microarc oxidation method (MAO) at various voltages with an electrolyte solution doped with DE particles. The coating exhibits excellent physical, electrochemical, and mechanical properties compared to those of the initial magnesium alloy. The corrosion resistance of the coated samples was approximately three orders of magnitude higher than that of the initial Mg alloy sample. Furthermore, the MAO coating significantly reduced the in vitro cytotoxicity in NIH/3T3 cells. This makes it an efficient biogenic tool for increasing the biocompatibility of Mg implants [71]. The surface of a biodegradable Mg alloy was modified to create porous DE biocoatings using the method of MAO. The effect of the addition of zirconium dioxide (ZrO_2_) microparticles on the structure and properties of DE-based protective coatings for Mg implants was studied. The coatings exhibited a porous structure and contained ZrO_2_ particles, with pores mostly less than 1 μm in size. Increasing the voltage in the MAO process increased the number of relatively larger pores (5–10 μm in size). Incorporating ZrO_2_ particles significantly affected the properties of DE-based coatings, with adhesive strength increasing by approximately 30% and corrosion resistance increasing by two orders of magnitude compared to coatings without zirconia particles [70].

The growing interest in the use of diatoms or DE in bone regeneration applications is due to changes in bone graft applications and markets. Diatoms and DE are primarily used as a supplement in bone regeneration and have been extensively studied for determining biocompatibility. Additionally, other biosilica-like sponge spicules are being studied for their potential applications in bone regeneration. In contrast, DE is more readily accessible and has a more established research base.

### 4.2. Wound Healing

Generally, wound healing requires an environment similar to the natural structure of the skin. Long-term wound healing can result in secondary complications, such as infection, pain, and discomfort during the healing process [73]. Therefore, wounds should be closed early. The wound dressing used differs based on the type of wound: hydrogels [74], films [75], and foams [76]. Proper selection and use of dressings are crucial for efficient wound healing as it reduces complications, accelerates the healing process, and decreases scarring even after the wound has fully healed [77,78] (Table 3). Thus, the preferred dressing should create a moist environment at the wound site, provide protection against bacterial invasion, and promote the active function of wound cells.

Diatom biosilica (DB) loaded with doxycycline (DOXY) has been developed and coated with hydroxybutyl chitosan (HBC) hydrogel to promote wound healing. The composite hydrogel HBC/DB/DOXY significantly inhibits the growth of *Staphylococcus aureus* and *Escherichia coli*. Moreover, the HBC/DB/DOXY hydrogel exhibits minimum cytotoxicity in L929 cells in vitro, thereby indicating excellent biocompatibility of this composite hydrogel. In vivo experiments showed that the HBC/DB/DOXY composite hydrogel enhances wound re-epithelialization and hastens the healing process. Wound closure area was assessed to be 99.4 ± 0.4% on the twelfth day following treatment with the hydrogel, accompanied by neovascularization and collagen deposition. These findings are indicative of the robust wound-healing abilities of the HBC/DB/DOXY hydrogel [79].

Rapid hemorrhage control is of great importance in both military trauma and traffic accidents. One study focused on the development of an effective and safe hemostatic composite sponge (AC-DB sponge) for controlling bleeding by combining alkylated chitosan (AC) and DB. The AC-DB sponge demonstrates rapid hemostatic capacity in vitro owing to its procoagulant chemical composition (with clotting time shortened by 78% compared to that in the control group), and it shows favorable biocompatibility (hemolysis ratio <5% and no cytotoxicity). The strong interaction between the AC-DB sponge and blood induced the activation, deformation, and aggregation of erythrocytes and platelets as well as intrinsic coagulation pathway activation, subsequently leading to significant coagulation acceleration. The AC-DB sponge exhibited exceptional performance during in vivo evaluation, demonstrating the shortest clotting time of only 106.2 s and minimal blood loss of only 328.5 mg. Therefore, the AC-DB sponge represents a safe and rapid hemostatic material with great potential in wound healing [80].

Considering the increase in drug resistance owing to the misuse of antibiotics, the development of non-conventional antibiotics for treating bacterial infections is necessary and urgent. A copper-deposited DB (Cu-DB) was developed using a hydrothermal method for treating infected wounds. Depositing Cu onto Cu-DB enhances the photothermal and photodynamic performance of DB, thereby increasing ROS levels during 808 nm laser irradiation. The light-capture effect of DB results in a stronger photothermal performance of Cu-DB than that of Cu-NPs, which significantly enhances the sterilization effect of Cu-DB. Cu-DB (6:5) almost completely inhibits *S. aureus* and *E. coli* growth within 10 min owing to the synergistic effect of photothermal and photodynamic activity. Trials conducted in vivo confirmed the antibacterial activity of Cu-DB for treating infected wounds and demonstrated their ability to expedite wound healing by suppressing inflammation, promoting collagen deposition, regulating the ratio of collagen type III to collagen type I, and promoting angiogenesis. Therefore, Cu-DB exhibits improved photothermal and photodynamic performance, and it has significant potential in treating infected wounds [81].

**Table 3 ijms-25-02023-t003:** Wound-healing application of diatom/diatomite biosilica.

Application	Type	Function	Form	Ref.
Wound healing	Diatomite	Promoting bioactivity of wound dressings for tissue regeneration	Scaffolds	[78]
	Diatom (*Cyclotella cryptica* sp.)	Biocompatibility, sustained drug release, non-adherence, and antibacterial activity with hemostatic properties	Hydrogel	[79]
	Silica nanoparticles (Diatom)	Accelerates diabetic wound healing	Hydrogel	[82]
Wound healing and hemostasis	Diatomite	Stops bleeding	Scaffolds	[83]
Biocompatibility and hemostasis	Diatomite	New hemostatic substance	Particles	[84]
	Diatomite	Hemostatic material with non-toxic side effects and rapid coagulation promotion	Particles	[85]
Hemostasis	Diatomite	Fast hemostasis with controlled porous structure	Aerogel	[86]
	Diatom (*Thalassiosira weissflogii, Thalassiosira* sp., *Cyclotella cryptica*)	Hemostasis and rapid blood clotting	Frustum	[87]
	Diatom (*Cyclotella cryptica* sp.)	Improves hemostasis efficiency	Frustule	[88]
	Diatomite	Hemostatic and antibacterial material	Spheres	[89]
Mechanical properties and hemostasis	Diatomite	Low-cost, high-efficiency, and rapid hemostasis material	Sponge	[90]
Antibacterial, hemostatic, and osteogenic	Diatomite	Bio-multifunctional sponge after tooth extraction	Sponge	[91]
Antibacterial	Diatomite	Antibacterial activity	Membrane	[92]
	Diatom (*C. cryptica*)	Healing of infected wounds, and suppressing inflammation, collagen, and angiogenesis	Particles	[81]

Several studies have evaluated the implementation of diatomite biosilica as a wound dressing material. This substance has the ability to create a suitable environment for the regeneration of skin via the absorption of exudate. Moreover, it can encourage skin regeneration by absorbing drugs or other substances that aid in skin repair in a direct or indirect manner. Notably, its biocompatibility and porosity make it an appropriate material for wound coverage.

### 4.3. Drug Delivery Systems

Currently, biosilica is being developed as an efficient drug delivery system (DDS) for use in various in vivo applications [53]. Previous biosilica-based DDS studies have mainly attempted to enhance the solubility of hydrophobic drugs. Nonetheless, recent research on biosilica has illustrated that the surface of this silica-based carrier can be modified for accumulating both hydrophobic and hydrophilic drugs [93].

Aside from its extensive utilization in diverse areas, researchers are exploring the use of diatom frustules in various fields [12,13]. Several smart nanostructured materials have been synthesized using chemical and biochemical methods that integrate, modify, transform, or mimic the biosilica shells of diatoms without altering their structural properties [53,94]. Further, diatom NPs have been obtained using mechanical crushing, sonication of diatom microfrustules, and acid solution purification. Diatoms can also be modified inorganically to transform them into a useful material for sustained drug release. Based on the nitrogen adsorption/desorption results, it was determined that diatom frustules possess a hollow surface area of 18.5 ± 0.8 m^2^ g^−1^, which is sufficient for the accumulation of a large quantity of drug molecules [95].

Recent studies have shown promising outcomes of using hybrid porous silica NPs as theragnostic nanodevices for imaging and drug delivery. Terracciano et al. proposed the use of hybrid gold-DE NPs as multifunctional devices for both imaging (photoacoustic and X-ray imaging) and drug delivery purposes [96]. Natural silica-based NPs were silanized with amino-propyltriethoxysilane (APTES) and modified with siRNA/poly-D-Arginine peptide to target the antiapoptotic factor B-cell lymphoma/leukemia-2 (Bcl2). The efficacy of *BCL2*-targeting siRNA-modified DE in decreasing gene expression was evaluated through quantitative real-time polymerase chain reaction and Western blot analysis. The observed gene silencing holds major biological importance and enables new possibilities for personalized lymphoma treatment using DE as a nanocarrier. Rea et al. explored the internalization kinetics and intracellular spatial distribution of non-targeting siRNA-loaded porous biosilica nanoparticles obtained from diatomite in H1355 lung cancer cells up to 72 h using label-free Raman spectroscopy [97].

Several studies have explored the modification of surfaces using hydrophobic (indomethacin) and hydrophilic (gentamicin) drugs. The presence of hydrophilic surfaces such as epoxy-rich 3-glycidoxypropyltrimethoxysilane and amino-rich APTES enhances the loading of hydrophobic drugs accompanied by a significant controlled release of the drug [98]. Organosilanes help control the hydrophilic and hydrophobic properties of diatom surfaces. A previous study study found that hydrophilic surfaces are more conducive to enhancing drug loading and controlling drug release than hydrophobic surfaces [99]. The interaction mechanism involving hydrogen bonding can be explained by the presence of large amounts of hydroxyl groups along with hydrophobic interactions and π–π linkage. These same factors have also been considered responsible for the sustained drug release observed in diatoms modified with graphene oxide (GO). The modification with GO improves the interaction between drug molecules and the layers of GO [100]. Although previous applications of biosilica have relied on fossil sources, other studies have shifted focus to culture-derived biosilica [101,102].

The use of diatoms or diatom-inspired/mimetic encapsulating or skeletonizing bodies as carriers for drugs and biomedical compounds has been explored. In addition, diatom biosilica has been used as a drug delivery vehicle through surface modification or coating with heterogeneous materials [103,104,105,106]. The surfaces of natural diatoms (DE) were modified with bioinspired polydopamine (PDA) to create a barrier/gatekeeper for the model drug curcumin (CUR). Subsequently, PDA was coated onto DE-CUR and tagged with the target ligand folic acid (FA) for use in targeted drug delivery applications. In vitro drug release studies revealed that at pH 1.2, 87.67% of the drug was released for pristine DE-CUR and 74.82% for DE-CUR-PDA-FA. At pH 7.4, DE-CUR released 67.53% of the drug, while DE-CUR-PDA-FA released 51.90%. The decrease in drug release can be attributed to the thin PDA layer coated on DE, which acts as a barrier or gatekeeper, controlling the drug release rate [107].

A recent study reported the potential application of diatom frustules in biomedical fields as their structure can be modified for various therapeutic purposes. Silica functionalization has been used to suppress cancer progression by delivering water-insoluble antitumor drugs. Additionally, DE particles coated with vitamin B12 facilitate the delivery of cisplatin and 5-fluorouracil, which are anticancer agents used to treat colorectal cancer [108].

### 4.4. Other Applications

Skin-attachable wearable devices, such as triboelectric nanogenerators (TENGs) and self-powered sensors, may present biocompatibility issues owing to their direct interaction with the human body. To address this issue, a chitosan-diatom TENG has been developed using biocompatible ocean-derived biomaterials, including diatom frustules and chitosan. The developed device is biocompatible and can be used in skin-attachable wearable applications. Diatom frustules, which are highly porous, naturally abundant, and mass-producible from ocean environments, can be used as a biocompatible additive to significantly alter the electro-positivity and surface properties of chitosan films. This alteration emphasizes the bioaffinity and delivers a significantly higher output power. The time-averaged power density of the chitosan-diatom TENG measures 15.7 mW/m^2^, which is 3.7-fold greater than that of a pure chitosan-based TENG. For practical purposes, the chitosan-diatom film with electrodes has been used in a self-powered wristwatch and a motion sensor that can be attached to the skin [109]. Additionally, DB has been used to amplify the output capabilities of TENGs based on cellulose nanofibrils (CNFs). The implementation of the diatom frustule-CNF TENG was evaluated by creating a self-powered intelligent mask for monitoring human respiration. While biomaterial-based TENGs (bio-TENGs) have been developed for wearable electronics and implantable sensors, they currently do not meet the basic requirements for practical applications. Therefore, a previous study has reported on the use of diatom frustules with amine and fluorine chemical functionalization to greatly enhance the output performance of bio-TENGs [110,111].

DE has been used as a new composite material in chitosan films, thereby improving their physicochemical properties [112]. The films display homogeneous blister-shaped structures formed by the inclusion of DE. The overall antioxidant activity of the composite films remains unaffected by the incorporation of DE, which can be attributed to the difficulty in forming radicals. Chitosan film incorporated with an increasing amount of DE demonstrates a significant improvement in antimicrobial activity. Furthermore, a previous study determined the potential interactions between chitosan and DE for the first time while using quantum chemical calculations [113].

Diatoms have been reported to be applied in chemotherapy. Curcumin-loaded magnetically active diatom frustules have been used in chemotherapy. The diatoms were cultured, and frustules were obtained via chemical and thermal processes. The frustules were rendered magnetically active through incorporation of iron oxide NPs using two different methods involving ferrofluid (CMDM-F) and in situ synthesis (CMDM-I) of iron oxide NPs for curcumin delivery. The obtained results suggest that diatoms can function as a unique platform for the delivery of drugs. A previous study has suggested that the developed CMDM can be used as a potential carrier to deliver cargo for efficient chemotherapy [114]. In a recent study, a combination therapy involving chemotherapy and photodynamic therapy (PDT) was evaluated. The researchers synthesized DE mesoporous silica NPs (dMSNs) with lanthanide metal ions in a PDT composition. The NPs generated ROS under near-infrared light irradiation and had magnetic resonance imaging and fluorescence imaging capabilities. Fucoidan extracted from *Sargassum oligocystum* in Pingtung Haikou exhibits the highest anticancer activity. For combination therapy, fucoidan has been incorporated into NPs and referred to as dMSN-EuGd@fucoidan. When applied to HCT116 cancer cells, the viability of cells treated with a concentration of 200 μg/mL was 57.4%. Cell viability experiments showed that dMSN-EuGd@fucoidan-treated cells had a cell viability rate of 47.7% when used at 200 μg/mL. Combination therapy exhibits higher anticancer efficacy than that obtained using either PDT or chemotherapy alone. NPs have been successfully synthesized to enable combined chemotherapy and PDT [115].

## 5. Challenges and Future Perspectives

Varius studies have discussed the desirable nature and potential versatility of diatoms [13,53,94]. Diatoms represent suitable materials for use as renewable resources to reduce carbon emissions. Nonetheless, more advancements are required in the commercial application of diatoms [116,117]. Hence, it is a crucial area of research with benefits to humans. In this review, we presented research from various groups evaluating the efficiency of diatom-derived silica for biomedical purposes. Previous studies have demonstrated that diatom biosilica can function as a drug transporter, a scaffold for regenerative medicine, and a material for in vivo enzyme immobilization [94]. We summarized the primary advantages and disadvantages associated with this biomaterial and conclude that the benefits outweigh any limitations. Future research should investigate the long-term compatibility of biosilica with biological systems while also exploring cost-effective and environmentally friendly methods for producing biosilica.

Notably, the exploitation of silica-based diatom cell walls for biotechnological applications implies that naturally abundant fossil sources (DE) can be considered a viable and cost-effective alternative to synthetic silica, which requires time-consuming and hazardous processes for its production. Overall, fossil biosilica sources can function as heavy metal sorbents, food-grade additives, fertilizers, or biosensors [118,119]. Living biosilica derived from diatoms may be more suitable for biomedical applications like drug carriers owing to the need for constant and known drug-release rates, which requires minimal variation in size, shape, and porosity of biosilica particles. Biosilica obtained from monospecific cultures possesses a predictable and uniform structure in contrast to fossil source-derived silica, which exhibits a range of morphological characteristics resulting in heterogeneity in frustule size, shape, and porosity. DE may be used as a source of silica when it is difficult to obtain diatom biosilica via a culture system.

To enhance the economic viability of large-scale diatom cultivation for biosilica production, other fractions of algal biomass can be exploited. Diatoms can function as photosynthetic biorefineries for several industrial processes, and the use of both the organic and inorganic fractions of algal biomass can aid in waste reduction [120]. The extraction of biochemical components from microalgal biomass does not impact the structure and integrity of silica frustules, which are unaffected even in acidic conditions. Culturing large diatoms can enable the simultaneous production of biosilica for biomedical purposes as well as the production of highly valuable organic compounds [117,121,122].

Taking into consideration the aforementioned findings, the use of diatom biosilica in the field of biomedical applications has increased significantly owing to its unique characteristics. To ensure the suitability of diatom biosilica for such applications, it is important to consider the purification, surface modification, and structural morphology. Although DDSs are currently well researched, they are being developed in a convergent manner in another biomedical application to create high value-added products in the future. From this perspective, it is necessary to conduct more diverse and yet convergent research for developments to be made.

## 6. Conclusions

Diatoms possess intricate 3D biosilica porous structures that are considerably more advanced and complex than fabricated porous materials which are expensive. Characterized by hierarchical pore structure, large surface area, easily modifiable surface chemistry, good permeability, non-toxicity, and high biocompatibility, diatom frustules have been exploited as low-cost materials for the preparation of innovative devices for various biomedical applications. The present review article outlines the applications of diatom biosilica-based scaffolds and composites in biomedical fields. We also evaluated the latest research in the biomedical field, including bone regeneration and wound dressings. The preparation of diatom platforms, surface chemical modifications, biocompatibility tests, cellular uptake, and drug loading/release capability were discussed. The results presented emphasize the advantages of using diatom biosilica as an inexpensive alternative to synthetic porous silica for the preparation of promising biomaterials.

## Figures and Tables

**Figure 1 ijms-25-02023-f001:**
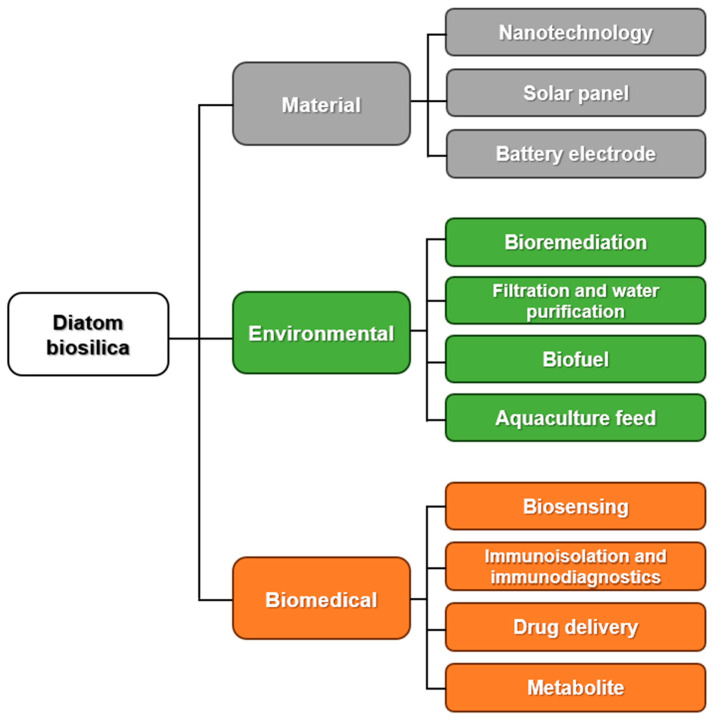
Schematic representation of the applications of diatom biosilica in various fields. The three fields are divided into sub-applications.

**Figure 2 ijms-25-02023-f002:**
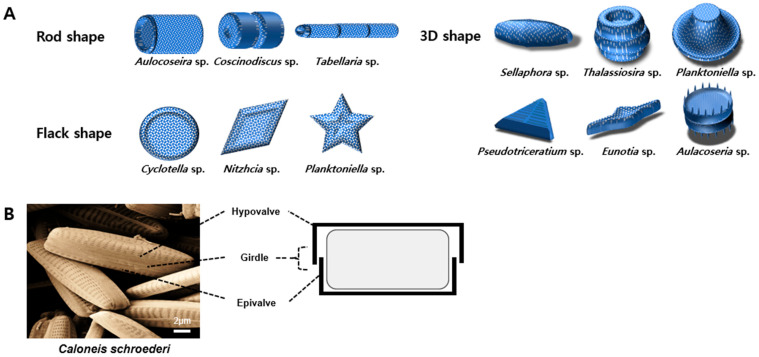
Schematic of the (**A**) unique structures of diatom frustules and (**B**) structure of *Caloneis schroederi* diatom frustules. 3D, three-dimensional.

**Figure 3 ijms-25-02023-f003:**
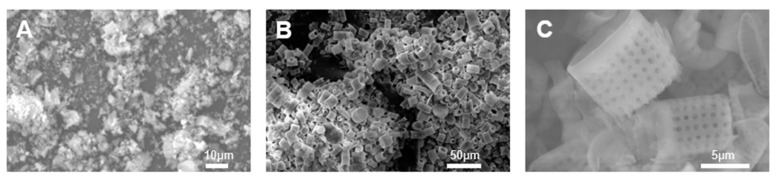
Image of (**A**) raw diatomite frustules, (**B**) structure in form of microcapsules generated after purification of raw diatomite, and (**C**) the pore structure of diatom frustules determined through scanning electron microscopy.

**Table 1 ijms-25-02023-t001:** Comparison of treatment methods used for diatom frustule cleaning.

Treatment		Advantages	Limitations	References
Baking	High temperature (calcination)	Reduction in use of hazardous chemicals	Possible alteration of pore size and possible post-treatments with acid solutions	[42]
Oxidation	H_2_SO_4_	High efficiency in organic matter removal	Hazardous chemical use, dissolution of thin frustules, and time-consuming post treatments	[43,44]
	H_2_SO_4_ + PTFE filters	Reduction in amount of acid required	Unsuitable for thin frustules	[45]
	HNO_3_	High efficiency in organic matter removal	High temperature treatments needed to increase efficiency	[44]
	Piranha solution (H_2_SO_4_ + H_2_O_2_)	High efficiency in organic matter removal	Time-consuming Post-treatments	[46]
	H_2_O_2_	Less dangerous than use of strong acids	Long incubation, and high temperature post-treatments needed to increase efficiency	[36]
	HCl	High purity of frustules	Possible frustule erosion depending on acid strength	[47]

PTFE, polytetrafluoroethylene.

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
