# Peer review of "Biomimetic Diatom Biosilica and Its Potential for Biomedical Applications and Prospects: A Review"

_ijms, 2024, doi:10.3390/ijms25042023_

Round 1

Reviewer 1 Report

Comments and Suggestions for Authors

Overall the review is written well and could be published. There are a few shortcomings that would need to be addressed. These are as follows:

Key papers are missing in the sections related to diatoms and bone growth, cell growth and regeneration. These include:

Adams et al.

Bioactive glass 45S5 from diatom biosilica

Journal of Science: Advanced Materials and Devices

Volume 2, Issue 4, December 2017, Pages 476-482

Han et al.

A comparison of the degradation behaviour of 3D printed PDLGA scaffolds incorporating bioglass or biosilica

Materials Science and Engineering: C

Volume 120, January 2021, 111755

Amoda et al.

Sintered nanoporous biosilica diatom frustules as high efficiency cell-growth and bone-mineralisation platforms

Materials Today Communications

Volume 24, September 2020, 100923

Hertz et al.

Preparation and characterisation of porous silica and silica/titania monoliths for potential use in bone replacement

Microporous and Mesoporous Materials

Volume 156, 1 July 2012, Pages 51-61

The authors should include a section at the beginning that discusses other reviews in this area and on how the current review that the authors propose is different. It is important that the readers know what it new that they get from this current review, and that seems to be lacking. Example reviews that could be included are:

Bayu et al.

Chapter 14 - Hierarchical nanoporous silica-based materials from marine diatoms

Handbook of Greener Synthesis of Nanomaterials and Compounds

Volume 2: Synthesis At the Macroscale and Nanoscale

2021, Pages 307-328

Sardo et al.

Mini-Review: Potential of Diatom-Derived Silica for Biomedical Applications

Appl. Sci. 2021, 11(10), 4533

(this is already cited - but should be included when comparing different reviews on the topic)

Maher et al.

Diatom Silica for Biomedical Applications: Recent Progress and Advances

Advanced Healthcare Materials

Volume7, Issue19

October 10, 2018

1800552

(this is already cited - but should be included when comparing different reviews on the topic)

Tramontano et al.

Nanostructured Biosilica of Diatoms: From Water World to Biomedical Applications

Appl. Sci. 2020, 10(19), 6811

Payares et al.

Microalgae Applications to Bone Repairing Processes: A Review

ACS Biomater. Sci. Eng. 2023, 9, 6, 2991–3009

Wang et al.

The Deep-Sea Natural Products, Biogenic Polyphosphate (Bio-PolyP) and Biogenic Silica (Bio-Silica), as Biomimetic Scaffolds for Bone Tissue Engineering: Fabrication of a Morphogenetically-Active Polymer

Mar. Drugs 2013, 11(3), 718-746

The use of diatoms or diatom inspired or diatom mimetic encapulating or skeletonizing bodies as carriers for drugs and biomedical compunds should be more thoroughly covered, examples are:

Aw et al.

Silica microcapsules from diatoms as new carrier for delivery of therapeutics

NANOMEDICINEVOL. 6, NO. 7

Uthappa et al.

Facile green synthetic approach of bio inspired polydopamine coated diatoms as a drug vehicle for controlled drug release and active catalyst for dye degradation

Microporous and Mesoporous Materials

Volume 288, 1 November 2019, 109572

Veliz et al.

Diatom-inspired skeletonisation of insulin – Mechanistic insights into crystallisation and extracellular bioactivity

Colloids and Surfaces B: Biointerfaces

Volume 133, 1 September 2015, Pages 140-147

Zobi F

Diatom Biosilica in Targeted Drug Delivery and Biosensing Applications: Recent Studies

Micro 2022, 2(2), 342-360

Wang and Guo

Bio-inspired encapsulation and functionalization of living cells with artificial shells

Colloids and Surfaces B: Biointerfaces

Volume 113, 1 January 2014, Pages 483-500

Gnanamoorthy et al.

Natural nanoporous silica frustules from marine diatom as a biocarrier for drug delivery. 

J Porous Mater 21, 789–796 (2014).

Delasoie and Zobi

Natural Diatom Biosilica as Microshuttles in Drug Delivery Systems

Pharmaceutics 2019, 11(10), 537

I think the above list contains some of the more important papers in the area, but if the authors wish to expand further from the above list, that would of course be welcomed as the review needs some additional depth that it is currently lacking.

Comments on the Quality of English Language

overall the English is fine and there is only minor proof reading that is needed

Author Response

# Overall the review is written well and could be published. There are a few shortcomings that would need to be addressed. These are as follows:
Key papers are missing in the sections related to diatoms and bone growth, cell growth and regeneration. These include:

(1) Adams et al.
Bioactive glass 45S5 from diatom biosilica
Journal of Science: Advanced Materials and Devices
Volume 2, Issue 4, December 2017, Pages 476-482

(2) Han et al.
A comparison of the degradation behaviour of 3D printed PDLGA scaffolds incorporating bioglass or biosilica
Materials Science and Engineering: C
Volume 120, January 2021, 111755

(3) Amoda et al.
Sintered nanoporous biosilica diatom frustules as high efficiency cell-growth and bone-mineralisation platforms
Materials Today Communications
Volume 24, September 2020, 100923

(4) Hertz et al.
Preparation and characterisation of porous silica and silica/titania monoliths for potential use in bone replacement
Microporous and Mesoporous Materials
Volume 156, 1 July 2012, Pages 51-61

- Answer: Thank you for your comment. We added content and edited.

# The authors should include a section at the beginning that discusses other reviews in this area and on how the current review that the authors propose is different. It is important that the readers know what it new that they get from this current review, and that seems to be lacking. Example reviews that could be included are:

(1) Bayu et al.
Chapter 14 - Hierarchical nanoporous silica-based materials from marine diatoms
Handbook of Greener Synthesis of Nanomaterials and Compounds
Volume 2: Synthesis At the Macroscale and Nanoscale
2021, Pages 307-328

(2) Sardo et al.
Mini-Review: Potential of Diatom-Derived Silica for Biomedical Applications
Appl. Sci. 2021, 11(10), 4533
(this is already cited - but should be included when comparing different reviews on the topic)

(3) Maher et al.
Diatom Silica for Biomedical Applications: Recent Progress and Advances
Advanced Healthcare Materials
Volume7, Issue19
October 10, 2018
1800552
(this is already cited - but should be included when comparing different reviews on the topic)

(4) Tramontano et al.
Nanostructured Biosilica of Diatoms: From Water World to Biomedical Applications
Appl. Sci. 2020, 10(19), 6811

(5) Payares et al.
Microalgae Applications to Bone Repairing Processes: A Review
ACS Biomater. Sci. Eng. 2023, 9, 6, 2991–3009

(6) Wang et al.
The Deep-Sea Natural Products, Biogenic Polyphosphate (Bio-PolyP) and Biogenic Silica (Bio-Silica), as Biomimetic Scaffolds for Bone Tissue Engineering: Fabrication of a Morphogenetically-Active Polymer
Mar. Drugs 2013, 11(3), 718-746

- Answer: Thank you for your comment. We added content and edited from line 72. We included the references as the reviewer requested. The relevant contents are as follow;

# The use of diatoms or diatom inspired or diatom mimetic encapulating or skeletonizing bodies as carriers for drugs and biomedical compunds should be more thoroughly covered, examples are:

(1) Aw et al.
Silica microcapsules from diatoms as new carrier for delivery of therapeutics
NANOMEDICINEVOL. 6, NO. 7 (2011)

(2) Uthappa et al.
Facile green synthetic approach of bio inspired polydopamine coated diatoms as a drug vehicle for controlled drug release and active catalyst for dye degradation
Microporous and Mesoporous Materials
Volume 288, 1 November 2019, 109572

(3) Veliz et al.
Diatom-inspired skeletonisation of insulin – Mechanistic insights into crystallisation and extracellular bioactivity
Colloids and Surfaces B: Biointerfaces
Volume 133, 1 September 2015, Pages 140-147

(4) Zobi Fet al.
Diatom Biosilica in Targeted Drug Delivery and Biosensing Applications: Recent Studies
Micro 2022, 2(2), 342-360 

(5) Wang and Guo et al.
Bio-inspired encapsulation and functionalization of living cells with artificial shells
Colloids and Surfaces B: Biointerfaces
Volume 113, 1 January 2014, Pages 483-500

(6) Gnanamoorthy et al.
Natural nanoporous silica frustules from marine diatom as a biocarrier for drug delivery. 
J Porous Mater 21, 789–796 (2014).

(7) Delasoie and Zobi
Natural Diatom Biosilica as Microshuttles in Drug Delivery Systems
Pharmaceutics 2019, 11(10), 537

 - Answer: Thank you for your comment. We added content and edited in line 372.

Reviewer 2 Report

Comments and Suggestions for Authors The article by Min et al is a review of existing publications on the topic Biomedical of applications of diatom biosilica. This topic is attractive and there are several reviews on it so far.   If we compare the review presented for consideration with existing reviews, for example, Tramontano et al. Nanostructured Biosilica of Diatoms: From Water World to Biomedical Applications // Appl. Sci. 2020, 10, 6811; doi:10.3390/app10196811; Rabiee et al. Diatoms with Invaluable Applications in Nanotechnology, Biotechnology, and Biomedicine: Recent Advances // ACS Biomater. Sci. Eng. 2021, 7, 3053−3068; Sardo et al. Mini-Review: Potential of Diatom-Derived Silica for Biomedical Applications // Appl. Sci. 2021, 11, 4533. https://doi.org/10.3390/app11104533, then it is inferior to them in the completeness of the material presented, the quality and accessibility of the presented drawings and diagrams. There are weaknesses in the text of the manuscript, which will be discussed below.

1.      The text of the manuscript contains sentences that need to be supported by references – Lines 24, 25, 27-28, 29, 92, 358, 429. 2.      Some references are inaccurate and do not reflect what is being said in the upcoming proposal – Lines 33, 74, 79, 83, 113. 3.      There are repetitions in the text. Lines 26-27 and 75-76. 4.      The content of Figure 1 is not clear. 5.      Fig. 2 inaccurate. The diatom shell shapes in Figure 2A are amazing, I would ask the authors to label the names of the species that have these special shell shapes. In Figure 2 B, the quality of the SEM photograph is not good enough and there is an error in the Girdle designation in the diagram on the right. 6.      Lines 189-193 should be included in chapter 4.3. 7.      There is no reference to Table 3 in the text. 8.      Line 149 - this is a reference to Figure 3, not 2. 9.      In Table 1, it is advisable to add horizontal lines, as in Table 2. 10.  In Table 3 why [[ ]]? 11.  Line 328 – ref. not inserted

The strength of this review is its novelty. It contains several links to articles from 2022-2023.

If the authors bring the quality of the text and illustrations to the level of MDPI journals, then I can recommend it for publication. I would reject it for now.

Author Response

The article by Min et al is a review of existing publications on the topic Biomedical of applications of diatom biosilica. This topic is attractive and there are several reviews on it so far.   If we compare the review presented for consideration with existing reviews, for example, Tramontano et al. Nanostructured Biosilica of Diatoms: From Water World to Biomedical Applications // Appl. Sci. 2020, 10, 6811; doi:10.3390/app10196811; Rabiee et al. Diatoms with Invaluable Applications in Nanotechnology, Biotechnology, and Biomedicine: Recent Advances // ACS Biomater. Sci. Eng. 2021, 7, 3053−3068; Sardo et al. Mini-Review: Potential of Diatom-Derived Silica for Biomedical Applications // Appl. Sci. 2021, 11, 4533. https://doi.org/10.3390/app11104533, then it is inferior to them in the completeness of the material presented, the quality and accessibility of the presented drawings and diagrams. There are weaknesses in the text of the manuscript, which will be discussed below.

1. The text of the manuscript contains sentences that need to be supported by references – Lines 24, 25, 27-28, 29, 92, 358, 429. 
- Answer: Thank you for your comment. We added references and edited. 

2. Some references are inaccurate and do not reflect what is being said in the upcoming proposal – Lines 33, 74, 79, 83, 113.
- Answer: Thank you for your comment. We edited them and checked.

3. There are repetitions in the text. Lines 26-27 and 75-76. 
- Answer: Thank you for your comment. We edited.

4. The content of Figure 1 is not clear.
- Answer: Yes, we agreed with your comment. We added the content of figure 1 in paragraph in line 64.

5. Fig. 2 inaccurate. The diatom shell shapes in Figure 2A are amazing, I would ask the authors to label the names of the species that have these special shell shapes. In Figure 2 B, the quality of the SEM photograph is not good enough and there is an error in the Girdle designation in the diagram on the right.
- Answer: Thank you for your comment. We edited content as you requested. 

6. Lines 189-193 should be included in chapter 4.3.
- Answer: Thank you for your comment. We edited content as you requested.

7. There is no reference to Table 3 in the text.
- Answer: We made it in mistake. We corrected them and edited.

8. Line 149 - this is a reference to Figure 3, not 2.
- Answer: We made it in mistake. We corrected them and edited.

9. In Table 1, it is advisable to add horizontal lines, as in Table 2.
 - Answer: Yes, we agreed with your comment. We edited as you requested.

10. In Table 3 why [[ ]]? 
- Answer: We made it in mistake. We corrected table 3.

11. Line 328 – ref. not inserted
- Answer: We made it in mistake. We edited. 

Reviewer 3 Report

Comments and Suggestions for Authors

The manuscript Biomimetic Diatom Biosilica and its Potential for Biomedical Applications and Prospectsdescribed by Ki Ha Min, Dong Hyun Kim, Sol Youn and Seung Pil Pack presents an interesting literature review of naturally occuring diatom biosilica as valuable materials for biomedical applications and may be accepted for publication in International Journal of Molecular Sciences.

 Remarks

Unfortunately, the Authors do not always use the terms diatoms and diatomite accurately in the text of the manuscript.

E.g. "Naturally occurring diatom microparticles contain impurities such as clay, volcanic, ....". In this case it should be "diatomite microparticles" but not diatom microparticles.

"Figure 3. .... (C) the pore structure of diatoms determined via scanning electron microscopy."  It need to change "the pore structure of diatoms" to "the pore structure of diatom frustules" because diatoms are live microalgae.

"It was reported that pure diatoms can be obtained using a pulverizer for diatom crushing or a piranha solution to remove impurities." Here we are also talking about diatomite but not living diatoms.

It is necessary to clearly distinguish diatoms (living microalgae) and diatomaceous earth (sedimentary rock consisting mainly of diatom biosilica fossils), biosilica obtained from living diatoms and biosilica of diatomaceous earth.

Author Response

The manuscript “Biomimetic Diatom Biosilica and its Potential for Biomedical Applications and Prospects” described by Ki Ha Min, Dong Hyun Kim, Sol Youn and Seung Pil Pack presents an interesting literature review of naturally occuring diatom biosilica as valuable materials for biomedical applications and may be accepted for publication in International Journal of Molecular Sciences.

 Remarks

Unfortunately, the Authors do not always use the terms diatoms and diatomite accurately in the text of the manuscript.

E.g. "Naturally occurring diatom microparticles contain impurities such as clay, volcanic, ....". In this case it should be "diatomite microparticles" but not diatom microparticles.
- Answer: Thank you for your comment. We edited.

"Figure 3. .... (C) the pore structure of diatoms determined via scanning electron microscopy."  It need to change "the pore structure of diatoms" to "the pore structure of diatom frustules" because diatoms are live microalgae.
- Answer: Thank you for your comment. We edited. 

"It was reported that pure diatoms can be obtained using a pulverizer for diatom crushing or a piranha solution to remove impurities." Here we are also talking about diatomite but not living diatoms.
- Answer: Thank you for your comment. We edited 내용 수정하기

It is necessary to clearly distinguish diatoms (living microalgae) and diatomaceous earth (sedimentary rock consisting mainly of diatom biosilica fossils), biosilica obtained from living diatoms and biosilica of diatomaceous earth.
 - Answer: Yes, we agreed with your comment. Edited to make it clearer, as you requested. 

Round 2

Reviewer 1 Report

Comments and Suggestions for Authors

The authors have incorporated the additional references well into the review and have also importantly, described to an extent, how their review is different to other reviews that currently exist.

Comments on the Quality of English Language

There are some minor grammatical issues that will be picked up by the typesetting team.